# LoRA-INT8 Whisper: A Low-Cost Cantonese Speech Recognition Framework for Edge Devices

**DOI:** 10.3390/s25175404

**Published:** 2025-09-01

**Authors:** Lusheng Zhang, Shie Wu, Zhongxun Wang

**Affiliations:** 1School of Physics and Electronic Information, Yantai University, Yantai 264005, China; ytdxeduzls@s.ytu.edu.cn (L.Z.); wushie@ytu.edu.cn (S.W.); 2Shandong Data Open Innovation Application Laboratory of Smart Grid Advanced Technology, Yantai University, Yantai 264005, China

**Keywords:** Cantonese speech recognition, Whisper, LoRA, INT8 quantization, parameter-efficient fine-tuning, edge computing

## Abstract

To address the triple bottlenecks of data scarcity, oversized models, and slow inference that hinder Cantonese automatic speech recognition (ASR) in low-resource and edge-deployment settings, this study proposes a cost-effective Cantonese ASR system based on LoRA fine-tuning and INT8 quantization. First, Whisper-tiny is parameter-efficiently fine-tuned on the Common Voice zh-HK training set using LoRA with rank = 8. Only 1.6% of the original weights are updated, reducing the character error rate (CER) from 49.5% to 11.1%, a performance close to full fine-tuning (10.3%), while cutting the training memory footprint and computational cost by approximately one order of magnitude. Next, the fine-tuned model is compressed into a 60 MB INT8 checkpoint via dynamic quantization in ONNX Runtime. On a MacBook Pro M1 Max CPU, the quantized model achieves an RTF = 0.20 (offline inference 5 × real-time) and 43% lower latency than the FP16 baseline; on an NVIDIA A10 GPU, it reaches RTF = 0.06, meeting the requirements of high-concurrency cloud services. Ablation studies confirm that the LoRA-INT8 configuration offers the best trade-off among accuracy, speed, and model size. Limitations include the absence of spontaneous-speech noise data, extreme-hardware validation, and adaptive LoRA structure optimization. Future work will incorporate large-scale self-supervised pre-training, tone-aware loss functions, AdaLoRA architecture search, and INT4/NPU quantization, and will establish an mJ/char energy–accuracy curve. The ultimate goal is to achieve CER ≤ 8%, RTF < 0.1, and mJ/char < 1 for low-power real-time Cantonese ASR in practical IoT scenarios.

## 1. Introduction

Cantonese, as one of the major Chinese dialects, is widely spoken in Guangdong, Hong Kong, and Macau. It serves not only as the primary medium of daily communication in these regions but also as an essential carrier of cultural heritage. Owing to its unique phonological features and deep cultural background, Cantonese holds significant academic value in linguistic and speech technology research. In particular, in the field of ASR [1], the nine lexical tones and rich phoneme inventory of Cantonese present diverse challenges and opportunities for speech model development.

However, achieving high accuracy in Cantonese speech recognition remains a significant challenge. Compared with Mandarin, the complexity of Cantonese syllables, phonemes, and tonal variations makes ASR systems perform less effectively, particularly under low-resource conditions. The scarcity of Cantonese speech data severely constrains the training of ASR systems; as a result, existing Cantonese ASR solutions often fail to obtain sufficient high-quality training data, leading to performance gaps compared with high-resource languages such as Mandarin. Although state-of-the-art models such as Whisper [2] and Wav2Vec2 [3] have achieved remarkable success on large-scale datasets in recent years, these models still struggle with accurate recognition of Cantonese syllables and tonal information.

Currently, research on Cantonese speech recognition primarily focuses on optimizing deep learning models, particularly the application of transfer learning to pretrained models such as Whisper and Wav2Vec2. By leveraging transfer learning, researchers can exploit the knowledge embedded in large-scale corpora to compensate for the scarcity of Cantonese data and improve recognition accuracy. However, despite the strong performance of these deep learning models on large-scale datasets [4,5], practical deployment—especially inference on edge devices—remains a considerable challenge. Edge devices typically have limited computational and storage resources, which makes deploying and running deep learning models, particularly ASR systems, extremely difficult. Traditional deep learning models often fail to achieve efficient, low-latency inference on edge platforms [6,7].

In addition, research on model quantization and deployment for edge devices remains relatively limited. Low-power edge computing devices, such as those in the Internet of Things (IoT), impose stringent constraints on computational and storage resources. How to deploy an efficient and accurate Cantonese ASR model in such resource-constrained environments remains an urgent challenge. Model quantization [8,9], as an effective technique for reducing computational complexity and storage requirements, has gradually become a research hotspot for optimizing edge deployment. Quantization can significantly lower the computational load and memory footprint of models, thereby enabling efficient inference for Cantonese ASR on resource-limited devices. However, due to the phonological complexity of Cantonese and the inherent variability of the data, achieving efficient quantization while preserving recognition accuracy remains a major open problem in current research.

In recent years, for ASR systems targeting low-resource languages, researchers have actively explored parameter-efficient tuning and quantization strategies based on pretrained models to improve deployment efficiency and recognition performance. Song et al. first proposed the LoRA-Whisper approach, which integrates low-rank adaptation (LoRA) modules into the Whisper model, effectively mitigating issues such as “language interference” and new language adaptation in multilingual recognition. Experiments covering eight languages reported relative improvements of 18.5% and 23.0% in multilingual recognition and language expansion tasks, respectively; however, this study did not specifically evaluate performance on low-resource languages such as Cantonese [10]. On the other hand, Andreyev systematically assessed the quantization performance of Whisper under INT4/INT5/INT8 configurations. On the LibriSpeech dataset, INT8 quantization reduced inference latency by approximately 19% and model size by 45% while maintaining recognition accuracy. Although these findings provide useful references for general quantization strategies, their applicability to low-resource or Cantonese scenarios remains unverified [11]. In addition, research on accent and speaker adaptation, such as MAS-LoRA and SAML, has introduced hybrid LoRA experts or speaker-adaptive mechanisms, achieving superior recognition performance on English accent and speaker datasets. However, these studies have primarily focused on English and have not addressed the challenges of low-resource languages like Cantonese [12].

Overall, although Cantonese ASR has made notable progress driven by deep learning and transfer learning, particularly demonstrating strong performance on large-scale datasets, it still faces multiple challenges under low-resource conditions. Addressing the inference efficiency of Cantonese ASR models on edge devices and integrating techniques such as model quantization to enable efficient, low-power deployment remain critical and challenging issues in current research.

In summary, although the aforementioned studies have provided valuable technical insights into LoRA-based fine-tuning, multilingual adaptation, quantization optimization, and even mixture-of-experts architectures, no prior research has systematically validated INT8 quantization on LoRA-fine-tuned Whisper models for Cantonese—a representative low-resource and tone-rich language. Comprehensive evaluations of the relationships among tonal recognition accuracy CER, and inference latency before and after quantization are still lacking, as are end-to-end experimental results under edge deployment conditions.

To address the aforementioned challenges, this paper proposes a cost-effective Cantonese ASR system based on LoRA-adapted Whisper, followed by INT8 quantization. The quantization process significantly reduces the model size and accelerates inference while demonstrating strong performance on edge devices, with improved energy efficiency and stable accuracy. CER evaluations confirm that quantization introduces minimal impact on tonal recognition, making the approach particularly suitable for tone-rich languages such as Cantonese.

## 2. Fundamental Principles

Currently, speech recognition has achieved remarkable progress on large-scale datasets; however, its deployment on edge devices still faces significant challenges. Edge devices typically have limited computational power and storage capacity, which restricts the direct application of complex models. Therefore, reducing model size and lowering computational requirements are key to enabling efficient edge deployment. By compressing model size, it is possible to not only substantially decrease storage demands but also accelerate inference, thereby improving the real-time performance and responsiveness of speech recognition systems on edge platforms.

### 2.1. Overview of the Whisper Model

Whisper is a multitask automatic speech recognition (ASR) system proposed by OpenAI, built on a Transformer-based architecture [13]. The model is trained on a large-scale speech dataset and supports multilingual and cross-task learning, including speech recognition and translation. Its primary goal is to achieve highly robust speech recognition across languages and low-resource conditions through large-scale supervised multilingual training. Unlike traditional methods that rely heavily on language-specific labeled corpora, Whisper leverages 680,000 h of paired audio–text data in its pretraining phase, covering 96 languages and diverse noise environments. This extensive pretraining enables Whisper to exhibit strong generalization capability and robustness in real-world applications.

Whisper employs large-scale supervised training within an end-to-end architecture to directly map speech signals to text sequences, eliminating the separate design of acoustic models, language models, and pronunciation lexicons commonly used in traditional ASR systems, thereby simplifying the overall system structure. Unlike self-supervised approaches such as Wav2Vec2, which focus on learning audio representations [14,15], Whisper adopts a fully supervised paradigm that directly optimizes the mapping from speech to text. This enables the model to learn joint representations of linguistic semantics and acoustic features, resulting in stronger adaptability across multiple tasks.

The overall architecture of Whisper adopts a typical sequence-to-sequence (Seq2Seq) structure based on an encoder–decoder Transformer design [16], which primarily consists of three components: a feature extraction module, a Transformer encoder, and a Transformer decoder. The feature extraction module first converts raw audio signals into 80-dimensional Mel spectrograms to capture spectral characteristics of speech. This representation is normalized along both time and frequency dimensions to enhance robustness under varying speech conditions. The encoder takes the Mel spectrogram as input and applies multiple layers of multi-head self-attention mechanisms and feed-forward networks to generate context-aware, high-dimensional speech feature representations:(1)Z=Encoder(X), Z∈ℝn×h,
where X denotes the input Mel spectrogram, n represents the number of frames, and h indicates the hidden dimension. The decoder adopts an autoregressive generation mechanism, predicting the next subword token based on previously generated tokens and the encoder output through a cross-attention mechanism:(2)Pyt | y<t , Z = Decodery<t, Z,

The final output is generated based on subword units, with special tokens introduced to control the task type (transcription/translation) and specify the target language. Whisper is trained entirely under a supervised learning paradigm using the cross-entropy loss function:(3)$$L_{\mathrm{ASR}}=-\sum_{t=1}^{T}\log P(y_{t}|y_{<t},Z),$$
where $y_{t}$ denotes the target token and T represents the length of the output sequence. Unlike self-supervised approaches, Whisper does not rely on masked prediction; instead, it performs direct sequence-to-sequence learning on paired speech–text data to ensure task consistency. The training corpus covers multiple languages and multiple tasks (ASR and speech translation) and includes a wide range of noisy scenarios, enabling the model to maintain robustness and transferability across multilingual and multi-domain tasks.

### 2.2. LoRA Method

With the widespread adoption of Transformer architectures in speech recognition, the number of model parameters has continued to increase, leading to substantial computational and storage demands during training and inference. For low-resource environments or edge devices, full fine-tuning not only incurs high computational costs but also requires storing multiple copies of the model, making deployment in practical scenarios challenging. To address this issue, LoRA (Low-Rank Adaptation) offers a parameter-efficient fine-tuning (PEFT) strategy that introduces only a small number of trainable parameters while maintaining model performance, enabling task-specific adaptation [17,18].

The core idea of LoRA is to freeze the pretrained model weights while introducing a low-rank decomposition as a trainable update to the weight matrices within the Transformer layers, thereby reducing the number of trainable parameters. Taking the linear transformation in the Transformer’s self-attention mechanism as an example, assume that the weight matrix in the pretrained model is given by:(4)W0∈ℝd×k,

In standard fine-tuning, the entire W0 needs to be updated, requiring d×k parameters. LoRA, however, decomposes the update term ΔW\Delta WΔW into the product of two low-rank matrices:(5)∆W=AB, A∈ℝd×r,B∈ℝr×k,r≪min(d,k),

Finally, the transformed weight can be expressed as:(6)$$W=W_{0}+∆W=W_{0}+AB,$$
where $W_{0}$ denotes the frozen pretrained weights, A,B represent the trainable low-rank matrices, and the rank r is typically set to 1–5% of the original dimension.

LoRA is typically applied to the attention projection layers and the linear transformation layers of the feed-forward network (FFN) in Transformer architectures. Its implementation mechanism involves two steps: first, freezing the pretrained weights to keep the original model parameters unchanged, which helps prevent overfitting and reduces storage overhead; second, introducing a low-rank module by inserting trainable low-rank matrices A,B into the specified linear layers to enable learnable updates. Finally, to stabilize training, LoRA introduces a scaling factor α to control the magnitude of the low-rank update:(7)$$\tilde{W}=W_{0}+\frac{\alpha }{r}AB,$$
for an input vector x, the linear transformation process in LoRA is formulated as:(8)$$\hat{h}=(W_{0}+AB)x=W_{0}x+A(Bx),$$
where $W_{0}x$ represents the frozen part, A(Bx) is the low-rank update, and the computational complexity is reduced from O(dk) to O(r(d + k)).

### 2.3. Model Quantization

When deploying the LoRA-adapted Whisper model on edge devices, model size and computational complexity remain major bottlenecks. Although LoRA significantly reduces the number of trainable parameters, the entire set of pretrained model weights still needs to be loaded during inference, resulting in high storage overhead and inference latency. For low-power devices such as IoT terminals and mobile platforms, further reducing model size and improving inference speed are critical. Model quantization, as an effective model compression technique, addresses this issue by reducing the numerical precision of parameters, thereby lowering computational cost and memory usage with minimal impact on accuracy. This makes quantization a key enabler for efficient edge deployment.

Quantization reduces storage requirements and accelerates matrix multiplication by mapping high-precision floating-point weights (e.g., FP32/FP16) to low-precision integers (e.g., INT8). Specifically, given the original floating-point weight wf∈ℝ, its quantized value wq is expressed as:(9)wq=roundwfs, s=max(|wf|)Qmax,
where s denotes the quantization scaling factor, and Qmax represents the range of integer representation (e.g., 127 for INT8). Dequantization restores an approximate floating-point value as w^f=s⋅wq. Through this scaling and reverse-scaling mechanism, quantization preserves the overall numerical distribution while reducing precision.

Quantization methods can be broadly categorized into three types. The first is static quantization, which calculates the activation distribution using a calibration dataset after training and generates quantization parameters offline. This approach is suitable for pre-deployment optimization. The second is dynamic quantization, which computes the activation scaling factors dynamically during inference, while weights are quantized at load time. This method requires no additional training or calibration and is ideal for rapid deployment of large models. The third is quantization-aware training (QAT), which simulates the effect of quantization during the training process. Although QAT provides the highest accuracy, it requires retraining and is therefore more costly.

The dynamic quantization mechanism applies separate strategies for weights and activations. For weight quantization, FP16 weights are converted to INT8 during model loading, with a fixed scaling factor s maintained. For activation quantization, scaling factors are computed dynamically during inference based on the input, which helps prevent quantization overflow.

For an input vector and its corresponding weight, the linear transformation in quantized inference can be expressed as:(10)$$y\approx (s_{w}w_{q})\cdot (s_{x}x_{q})=s_{w}s_{x}(w_{q}\cdot x_{q}),$$
where $w_{q},x_{q}$ are represented in INT8 format, and $s_{w},s_{x}$ denote the corresponding scaling factors.

### 2.4. Overall Model Architecture

In this study, we integrate Whisper, LoRA, and INT8 quantization to construct an efficient Cantonese ASR system. The overall architecture of the proposed model is illustrated in Figure 1.

In this architecture, the Whisper model serves as the backbone, providing multilingual speech recognition capabilities and effectively handling syllables and tonal variations in Cantonese. LoRA fine-tuning is then applied to the Whisper model, significantly reducing the number of trainable parameters and thereby lowering both training and inference costs. Finally, the model is quantized to INT8, enabling efficient execution on edge devices while maintaining recognition performance.

## 3. Training Procedure

### 3.1. Dataset and Preprocessing

The primary training data in this study is sourced from the Cantonese subset of the Common Voice dataset and the MDCC corpus [19]. Cantonese differs from Mandarin in terms of tonal and syllabic structure and contains a high degree of colloquial expressions in everyday communication. Therefore, selecting a Cantonese-specific corpus is essential for evaluating the robustness of the proposed method under real-world conditions characterized by multiple tones and variable pronunciations. All audio recordings were resampled to 16 kHz, and the text annotations were standardized in Traditional Chinese format.

To improve speech feature consistency and text normalization, the following processing steps were applied: (1) Deduplication and silence trimming: remove leading and trailing silence segments longer than 500 ms using the SoX silence function; (2) Mel-spectrogram extraction: generate 80-bin Mel-spectrogram with a window length of 25 ms and a frame shift of 10 ms, followed by dB normalization; (3) Text normalization: convert full-width characters to half-width; remove meaningless punctuation marks; retain numbers and Cantonese particles.

### 3.2. Training Configuration

In the training process of the speech model, the choice of optimizer directly affects the convergence speed and stability of the model. Based on this, this study adopts the AdamW optimizer to update model parameters [20]. AdamW combines the advantages of momentum and adaptive learning rate from the traditional Adam optimizer and introduces a weight decay mechanism, which enables more stable and efficient convergence on large-scale corpora and deep network structures. For hyperparameter configuration, the initial learning rate is set to 1 × 10^−3^ to ensure a fast initial training speed while avoiding excessive oscillation.

For the t-th iteration, the parameter update rule of AdamW is as follows:(11)$$m_{t}=\beta_{1}m_{t-1}+\left(1-\beta_{1}\right)g_{t},$$(12)$$v_{t}=\beta_{2}v_{t-1}+\left(1-\beta_{2}\right)g_{t}^{2}$$(13)$${\hat{m}}_{t}=\frac{m_{t}}{1-\beta_{1}^{t}},\;{\hat{v}}_{t}=\frac{v_{t}}{1-\beta_{2}^{t}},$$(14)$$\theta_{t}=\theta_{t-1}-\eta \cdot \frac{{\hat{m}}_{t}}{\sqrt{{\hat{v}}_{t}}+\epsilon }-\eta \cdot \lambda \theta_{t-1},$$

In parameter-efficient fine-tuning, LoRA modules are inserted into the projection layers of the Transformer attention mechanism and the linear mapping layers of the feed-forward network to achieve low-rank parameter updates. In this study, LoRA is applied to the Query projection (q_proj), Value projection (v_proj), and the first linear transformation (fc1) in the feed-forward sublayer. The q_proj and v_proj layers directly affect the computation of attention weights, which is critical for capturing contextual relationships, while the fc1 layer determines the transformation capability of the feed-forward network, significantly impacting the expressive power of speech feature modeling.

The LoRA hyperparameters are configured as follows: rank r = 8, scaling factor α = 32, and dropout probability = 0.05. The rank r defines the dimension of the low-rank update matrices; r = 8 is chosen to balance performance and parameter size. The scaling factor α controls the magnitude of the low-rank update and is set to 32 to prevent overfitting and accelerate convergence. A dropout rate of 0.05 introduces stochastic regularization, further improving the generalization ability of the model.

### 3.3. Dynamic Quantization Process

To further compress the model and reduce inference latency, this study employs INT8 dynamic quantization using ONNX Runtime without requiring retraining. First, the LoRA-fine-tuned Whisper model is exported to an ONNX graph while maintaining FP16 precision to balance numerical stability and quantization accuracy. Next, under the ONNX Runtime framework, both weights and activations are dynamically quantized by mapping floating-point representations (FP16) to low-precision integers (INT8). This approach adjusts scaling factors dynamically based on the input distribution, ensuring that quantization errors remain within a controllable range and maintaining model performance during inference. Finally, before quantization, LoRA weights are merged into the main weight matrices to avoid introducing additional operations during inference. Through this strategy, the final quantized model retains the same structure as the original model, enabling efficient loading and execution.

For a comprehensive comparison with existing quantization schemes, Table 1 provides a summary of representative methods with respect to training requirements, accuracy variation, latency optimization, and deployment complexity, while also highlighting the advantages and limitations of the dynamic quantization strategy employed in this study. The results indicate that dynamic quantization enables substantial latency reduction and model compression without necessitating retraining, thereby offering a practical and efficient solution for rapid edge deployment scenarios.

Supplementary Notes:

PTQ: Advantages—Fast deployment, simple implementation, and no need for retraining. Limitations—Sensitive to data distribution; may incur significant accuracy loss in tasks with high precision requirements.

QAT: Advantages—Provides the best post-quantization accuracy, often close to the original model. Limitations—Requires retraining, which is costly and complex to implement.

Dynamic Quantization: Advantages—Does not require retraining, offers good cross-platform compatibility, and achieves notable latency reduction. Limitations—Slightly less effective than QAT for activation quantization, and may be unstable under extremely high precision demands.

Dynamic Fixed-Point Quantization: Advantages—Hardware-friendly and effective in latency reduction. Limitations—Strongly hardware-dependent and requires adaptation to the target platform.

Activation Quantization: Advantages—Reduces memory usage with minimal impact on accuracy. Limitations—Limited effectiveness in latency optimization.

Weight Sharing Quantization: Advantages—Provides extremely high compression rates. Limitations—Causes substantial accuracy degradation, making it suitable only for tasks that are less sensitive to precision.

Learned Quantization: Advantages—Allows end-to-end optimization of quantization parameters, balancing accuracy and efficiency. Limitations—Involves a complex training process with high computational cost.

In this study, ONNX Runtime INT8 dynamic quantization was ultimately adopted, as it requires no retraining, maintains accuracy within a controllable range, and reduces the model size to 60 MB. These characteristics make it particularly suitable for scenarios that demand rapid deployment and efficient inference on resource-constrained edge devices.

### 3.4. Evaluation Metrics

The most commonly used evaluation metric in speech recognition tasks is error rate, which can be further categorized into phoneme error rate, character error rate (CER), word error rate (WER), and sentence error rate depending on the modeling unit. Considering that Cantonese exhibits a rich representation at the character level and the recognition units in this study are closer to character-based outputs, CER is chosen as the primary metric.

CER is calculated by comparing the recognized output with the reference text using edit distance, which includes substitution, insertion, and deletion operations. Its formula is:(15)CER=(S+D+I)/N,
where S, D and I denote the number of substitutions, deletions, and insertions, respectively, and N represents the total number of characters in the reference text.

## 4. Results and Analysis

### 4.1. ASR Accuracy

As shown in Table 2, the original Whisper-tiny model exhibited a CER as high as 49% on the test set, rendering it nearly unusable. After full fine-tuning, the CER dropped to 10.3%, indicating that with sufficient Cantonese data, large models can effectively learn tonal and syllabic patterns. LoRA fine-tuning, which updates only 1.6% of the parameters, achieved a CER only about 0.8 percentage points higher than full fine-tuning, while reducing memory usage for training (including weights, gradients, and optimizer states) and backward computation cost by approximately one order of magnitude (≈10×). Applying INT8 quantization to the LoRA-fine-tuned model compressed it to 60 MB, while maintaining a CER of about 12.6%, representing a 37 percentage point reduction compared to the original model and meeting practical accuracy requirements.

### 4.2. Decoding Efficiency

As shown in Table 3, INT8 quantization improves the real-time factor (RTF) on a CPU to achieve 5× real-time performance, reducing inference latency by 43%. On a GPU, the RTF reaches 0.06, enabling batch processing to support high-concurrency services at a rate of over 1500 ms of audio per second per card. Quantization introduces almost no additional memory overhead, making it suitable for edge devices or cost-efficient cloud deployment.

### 4.3. Ablation Study

As shown in Table 4, r = 8 achieves the best trade-off between accuracy and cost: it requires only about 1.6% of the parameters while delivering accuracy comparable to full fine-tuning, and inference speed remains unaffected even after quantization. Further reducing the rank to r = 4 decreases the number of parameters but increases CER by 1.3 percentage points, indicating that excessively low rank compromises the model’s expressive capacity. Combined with Table 2, it can be observed that LoRA + INT8 achieves the best balance across accuracy, speed, and model size, making it more suitable for edge deployment compared to full fine-tuning with FP16 precision.

### 4.4. Comparative Study

To validate the effectiveness of the proposed method, we compared it with publicly available Cantonese ASR systems. Table 5 summarizes the performance of our method and representative Cantonese ASR models in terms of accuracy, inference latency, and model size.

### 4.5. Limitations

Although this study has achieved certain progress, several limitations remain to be addressed in future work. First, the training data primarily rely on Common Voice and MDCC, with limited coverage of news broadcasts, long conversational sentences, and domain-specific terminology. Second, the evaluation was only partially conducted on edge hardware and did not include comprehensive testing on x86/ARM CPUs, nor verification on MCU/NPU platforms such as EdgeTPU and K210. Finally, the LoRA structure was fixed, with rank and insertion layers manually configured, without task-specific or model-size-specific optimization.

## 5. Future Work

To further advance the real-world deployment of Cantonese ASR, future work will focus on four dimensions: data, model, deployment, and evaluation. First, we plan to expand the training corpus by crawling and aligning Cantonese subtitles from movies and TV series, Hong Kong stock market announcements, and Cantonese podcasts to build a more comprehensive dataset. The current LoRA configuration, including rank and insertion layers, was manually set and may not be optimal for different models or tasks; therefore, we will explore adaptive approaches such as AdaLoRA or Auto-LoRA, leveraging Fisher information or SVD to automatically allocate rank across layers. We also intend to investigate the applicability of GPTQ-INT4 and SmoothQuant in Transformer-based ASR scenarios. On the deployment side, we will migrate the model to TFLite with NNAPI and EdgeTPU support to validate real-time performance, while introducing continuous power sampling using powermetrics, Intel RAPL, and ARM PMU. The evaluation will report energy–accuracy trade-offs, including mJ/char–accuracy curves and per-energy-unit character correctness.

Through a progressive upgrade incorporating large-scale self-supervised learning, tone-aware optimization, adaptive LoRA, extreme quantization, and streaming inference, combined with comprehensive energy and latency evaluation, future versions aim to achieve CER ≤ 8% under real-world noisy Cantonese speech, while delivering RTF < 0.1 and mJ/char < 1 on mobile SoCs and IoT NPUs, enabling low-power real-time Cantonese ASR for applications such as smart homes, in-car voice assistants, and wearable devices.

## Figures and Tables

**Figure 1 sensors-25-05404-f001:**
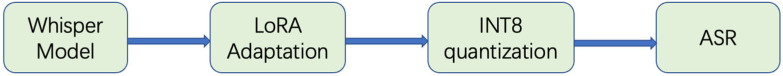
System Architecture.

**Table 1 sensors-25-05404-t001:** Comparative Analysis of Quantization Methods.

Method	Training Requirement	Accuracy Variation	Latency Optimization	Model Compression Ratio
PTQ	No retraining required	Slight drop (1–3%)	Medium	~2–4×
QAT	Retraining required	Minimal loss (<1%)	High	~2–4×
Dynamic Quantization	No retraining required	Slight drop (1–2%)	High (CPU speedup 30–50%)	~2–3×
Dynamic Fixed-Point	Partial retraining	Small accuracy drop	High	~2–3×
Activation Quantization	Optional retraining	Small accuracy drop	Medium–High	~2×
Weight Sharing	Retraining required	Relatively large accuracy drop	Medium	~4×+
Learned Quantization	Retraining required	Minimal accuracy loss	High	~2–4×

**Table 2 sensors-25-05404-t002:** ASR Accuracy.

Model	Model Size	CER (%)
Whisper-tiny (Original)	151 MB	49.47
Whisper-tiny	151 MB	10.29
LoRA-tiny	165 MB	11.13
LoRA-tiny INT8	60 MB	12.55

**Table 3 sensors-25-05404-t003:** Decoding Efficiency.

Model	Platform	RTF	Latency per 10 s of Audio
FP16-tiny	M1 Max CPU	0.35	3.5 s
INT8-tiny	M1 Max CPU	0.20	2.0 s
INT8-tiny	A10 GPU	0.06	0.6 s

**Table 4 sensors-25-05404-t004:** Ablation Study.

Configuration	Parameter Count	CER (%)	RTF
Full fine-tuning	+100%	10.29	0.25
LoRA (r = 8)	+1.6%	11.13	0.2
LoRA (r = 4)	+0.8%	12.41	0.2

**Table 5 sensors-25-05404-t005:** Comparative Study.

Method	Model Size	CER (%)	RTF
Proposed LoRA-INT8 Whisper-tiny	60 MB	12.55	0.20
Whisper-MCE [21]	155 MB	12.61	N/A
Fairseq S2T [19]	~310 MB	10.15	N/A
Wav2Vec2-Base (CV official baseline)	320 MB	14.6	N/A

Note: To the best of our knowledge, there are currently no published works on quantized Cantonese ASR models. Therefore, we primarily compare our method against full-precision Cantonese ASR systems, including Whisper-MCE (Xie et al., 2023) [21] and Fairseq S2T (Yu et al., 2022) [19], while additionally reporting latency and model size to highlight its deployment-oriented advantages. Results show that our method maintains competitive recognition accuracy while significantly outperforming existing models in terms of latency and size, making it suitable for low-resource and edge-device deployment.

## Data Availability

Restrictions apply to the availability of these data. Data are available at https://commonvoice.mozilla.org.

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
