# Peer review of "LoRA-INT8 Whisper: A Low-Cost Cantonese Speech Recognition Framework for Edge Devices"

_sensors, 2025, doi:10.3390/s25175404_

Round 1

Reviewer 1 Report

Comments and Suggestions for Authors

In this paper, the author proposes a cost-effective Cantonese ASR system based on LoRA fine-tuning and INT8 quantization to address the triple bottlenecks of data scarcity, oversized models, and slow inference that hinder Cantonese automatic speech recognition (ASR) in low-resource and edge-deployment environments.

When developing a speech recognition system, several key considerations must be addressed to ensure the system's effectiveness and efficiency. These include the choice of dynamic quantization method, evaluation metrics, dataset selection, and dataset composition.

1) Dynamic Quantization Methods:

There are several methods for Dynamic Quantization : Post-Training Quantization (PTQ), Quantization-Aware Training (QAT, Dynamic Fixed-Point Quantization, Activation Quantization, Weight-Sharing Quantization and  Learned Quantization.

  1. The author must justify the choice of their quatization method by comparing their method with their previous methods.
  2. The choice of quantization method should be justified by comparing the results with previous methods, highlighting the advantages and potential limitations of the chosen approach.

2) Evaluation Metrics:

There are several methods for evaluation metrics in speech recognition : Word Error Rate (WER), Character Error Rate (CER), Sentence Error Rate (SER),  Accuracy, Precision, Recall, F1-score, Real-Time Factor (RTF) and Latency.

  1. The author must measure these evaluators and justify their choice of evaluator
  2. The choice of evaluation metrics should be justified based on the specific requirements of the application, and the results should be presented in a way that allows for comparison with other methods.

3) Datasets:

There are two datasets : Cantonese Common Voice Dataset and Multi-Domain Cantonese Corpus (MDCC) :

  1. The choice of dataset should be justified based on the specific requirements of the application.
  2. The composition of the training, validation, and test sets should be clearly specified.
  3. The results should be presented using field-cross validation or as an average value with a standard deviation to ensure the reliability of the results.

4) comparative study

  1. a) To demonstrate the robustness of his proposed method, the author must compare his results with those published in the literature.

Reviewer 2 Report

Comments and Suggestions for Authors

The paper deals with an important problem of speech recognition of Cantonese language. In general, the Chinese language for non-Chinese people sounds extremely difficult to differentiate syllables, words, etc. The Cantonese is probably even more difficult. Thus, the findings presented in the article might be helpful for implementing automatic speech recognition systems. However, there can be pointed two issues that could be addressed by the authors to improve the general quality of the paper.

  1. The authors claim that their solution is particularly dedicated for edge devices. However, the experimental results of Table 2 were obtained using a platform with relatively high computing power, probably unreachable for most edge devices. It would be very interesting to see the results obtained from platforms that are much less resource-full. How many CPU cores were used in the case of the M1 Max CPU platform? And a second question – do any special software packages/libraries are needed for edge devices in order to implement a system from Fig. 1? Can this system be implemented in edge devices using Python, for example?
  2. Research papers usually contain a section which shows a comparison of the authors’ work with other similar solutions. This is not quite the case as far as this paper is concerned. Can the authors provide (if possible) such a comparison taking into account, for example, the CER, model size, or latency?

Reviewer 3 Report

Comments and Suggestions for Authors

The manuscript proposes a parameter-efficient adaptation of OpenAI's Whisper-tiny model for Cantonese ASR using Low-Rank Adaptation (LoRA) and INT8 dynamic quantization via ONNX Runtime. The aim is to enable low-latency, low-memory deployment on edge devices. Experiments show the model achieves performance close to full fine-tuning while significantly reducing size and inference time.

Despite the practical relevance, the manuscript lacks the level of novelty and theoretical contribution expected for journal publication. Specifically, there is no novel algorithmic or theoretical contribution (for example, there is no custom quantization design). The paper applies existing techniques (LoRA and INT8 quantization) in a standard way to a known model (Whisper-tiny). There is no new methodology or model innovation proposed. Also, the experiments are limited to standard datasets and a small hardware profile (M1 Max and one A10 GPU), which limits the relevance for broader edge device deployment (i.e. there is no discussion related to tiny IoT devices and possible practical applications).

Round 2

Reviewer 1 Report

Comments and Suggestions for Authors

The author has addressed all the issues raised in the revised version of the report. 

Reviewer 3 Report

Comments and Suggestions for Authors

The authors have improved the manuscript according to the comments.